# Exploring the Impact of Corpus Diversity on Financial Pretrained Language Models

**Jaeyoung Choe[1]***, **Keonwoong Noh[2]**, **Nayeon Kim[1]***, **Seyun Ahn[2]**, **Woohwan Jung[1,3]**

[1] Department of Applied Artificial Intelligence, Hanyang University
[2] Division of Computer Science, Hanyang University
[3] Ramply Inc.

{cjy9100, rohgw011, na2na8, tpdbs0907, whjung}@hanyang.ac.kr

## Abstract

Over the past few years, various domain-specific pretrained language models (PLMs) have been proposed and have outperformed general-domain PLMs in specialized areas such as biomedical, scientific, and clinical domains. In addition, financial PLMs have been studied because of the high economic impact of financial data analysis. However, we found that financial PLMs were not pretrained on sufficiently diverse financial data. This lack of diverse training data leads to a subpar generalization performance, resulting in general-purpose PLMs, including BERT, often outperforming financial PLMs on many downstream tasks. To address this issue, we collected a broad range of financial corpus and trained the **Fi**nancial **L**anguage **M**odel (FiLM) on these diverse datasets. Our experimental results confirm that FiLM outperforms not only existing financial PLMs but also general domain PLMs. Furthermore, we provide empirical evidence that this improvement can be achieved even for unseen corpus groups.

## 1 Introduction

Pretrained Language Models (PLMs) have been successfully employed in various natural language processing tasks. This trend has been extended to further pretraining of PLMs on domain-specific corpus in various fields such as biomedical (Lee et al., 2020), scientific (Beltagy et al., 2019), and clinical (Alsentzer et al., 2019) domains.

In financial domain, Araci (2019); Yang et al. (2020); Liu et al. (2021); Loukas et al. (2022) proposed domain-specific PLMs based on BERT (Devlin et al., 2019) model. However, each study tested a small number of tasks, resulting in inadequate validation. We conduct extensive experiments to show that the generalization performance of existing financial PLMs is unsatisfactory. Even general domain PLMs such as BERT and RoBERTa (Liu et al., 2019) outperformed the financial PLMs on financial tasks.

To investigate the reasons for the low generalization performance, we categorize the financial corpus into five groups and examine their use in existing financial PLMs. This reveals that most existing PLMs are pretrained on a corpus consisting of only two of these corpus groups. This indicates that the pretraining data for financial PLMs lacks diversity, which might cause their low generalization performance across various financial tasks.

Motivated by this observation, we collect a broad range of corpus and analyze the impact of corpus diversity on the performance of language models. Our investigations demonstrate that as a corpus becomes more diverse, the model's performance improves. Incorporating a diverse corpus, even with fewer tokens, yields a better generalization performance than relying on numerous tokens from a non-diverse corpus. Furthermore, when using diverse corpora, the model exhibits robust generalization performance on unseen tasks.

We train our **Fi**nancial **L**anguage **M**odel (FiLM) on a corpus with diverse documents and evaluate it on six financial tasks, including recently introduced works (Chen et al., 2021; Loukas et al., 2022; Shah et al., 2023). Our experimental results show that FiLM outperforms not only existing financial PLMs but also general domain PLMs on most financial tasks. In addition, we achieve an improvement of performance while reducing the number of tokens trained and energy consumption. To the best of our knowledge, FiLM is the first model to surpass RoBERTa in financial domain. We make our model and code publicly available on GitHub[1] and Huggingface hub[2] for continuous advancement in financial domain.

---

*Major in Bio Artificial Intelligence

[1] https://github.com/deep-over/FiLM
[2] https://huggingface.co/HYdsl/FiLM

| Group | Name | Description | Financial PLMs | # Tokens |
|---|---|---|---|---|
| News | TRC2 | Financial news stories from Reuters | i, iv | 227.3M |
| | Investing.com | Stock, options, commodity etc. News article | - | 130.8M |
| | NYtimes | Economic articles from the New York Times | - | 75M |
| | EIA | Commodity related news articles from EIA | - | 1.1M |
| SEC filings | | Annual reports(10-K) and quarterly reports(10-Q) | ii, iv, v | 307.1M |
| Earnings Call | | Earnings conference call transcripts | ii, iv | 1.6B |
| Papers | ArXiv | A collection of abstracts of economic research papers | - | 42.1M |
| | AIHUB | A collection of Korean economics research papers | - | 5.8M |
| MISC | Investopedia | Economic glossary | iv | 5.3M |
| | FinWEB | Finance, loans, and insurance related articles | iii | 2.8M |
| | | A total of 10 datasets | | 2.4 B |

Table 1: Summary of the financial datasets used for pre-training the models. The 'Financial PLMs' column represents the financial PLMs that have utilized each respective dataset: i. Araci (2019), ii. Yang et al. (2020), iii. Liu et al. (2021), iv. Shah et al. (2022), v. Loukas et al. (2022)

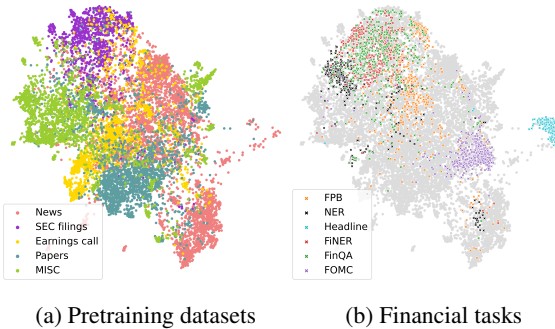

(a) Pretraining datasets    (b) Financial tasks

Figure 1: Sentence embeddings visualization for both corpus groups and financial tasks.

## 2 Proposed method

First, the datasets used to train the model are introduced. Second, we describe the method for pre-processing the datasets. Finally, we describe FiLM training procedure.

### 2.1 Pretraining datasets

For further pretraining the language models, we collected a financial corpus from 10 sources. There are various financial documents with different characteristics. For instance, financial reports have a higher prevalence of numerical information than other texts such as the abstracts of research in finance (Loukas et al., 2022).

As shown in Table 1, we categorize the pretraining datasets into the five groups as follows:

- News: The datasets are sourced from financial news articles.
- SEC filings: This dataset comprises financial reports (10-K, 10-Q) submitted to the U.S. Securities and Exchange Commission (SEC).
- Earnings call: This dataset comprises the information and transcripts of earnings confer-

ence calls obtained from Seeking Alpha.
- Papers: This group contains the abstracts of research papers in the field of economics.
- MISC: It includes other datasets in the financial domain.

For more detailed information on the corpus and groups, please refer to Appendix A.1. Figure 1a shows the embeddings of sentences in the pretraining corpus. The color of each point represents the group to which the corresponding sentence belongs. This indicates that sentences within the same group have similar embeddings, thereby forming clusters. For example, sentences in the financial news group are primarily distributed in the right and lower areas, whereas SEC filings are concentrated on the top side.

None of the corpus groups was distributed over the entire space. From this observation, we could conclude that it is essential to use data from all groups for pretraining to enable the language model to learn diverse features.

However, existing studies do not use all these groups to further train language models. In addition, most of them (Yang et al., 2020; Araci, 2019; Loukas et al., 2022) use only two groups of datasets. Consequently, these models might not achieve high performance across the entire financial domain. To attain robust performance over the entire financial domain, we employ all groups of datasets for pretraining.

### 2.2 Preprocessing

The preprocessing for the pretraining involves two steps: cleaning and deduplication. In the cleaning step, we remove HTML tags using Beautiful-Soup[3] and unnecessary special characters such as

---

[3] www.crummy.com/software/BeautifulSoup/

newline characters. Deduplication could improve the efficiency of pretraining and the accuracy of PLMs (Lee et al., 2022). Thus, we remove duplicate sentences from the datasets during preprocessing. For deduplication, we remove all but one sentence from each group of duplicate sentences in the corpus. The contents of the remaining documents were preserved if a sentence was removed during deduplication. After deduplication, the token count decreased from 3.3B to 2.4B, and the size of the training dataset reduced from 19.5GB to 14GB.

## 2.3 Training procedure for FiLM

To train FiLM, we further pretrain RoBERTa on financial corpus presented in Section §2.1. Following RoBERTa, we use the masked language model for the pretraining task. In addition, we use the same tokenization method and model input format as in RoBERTa. We further pretrain our model for only one epoch because the performance saturates after one epoch. We use a batch size of 16 and a learning rate of 1e-5. We use the Adam optimizer and do not set any specific scheduler or warm-up steps. Please refer to Table 5 for the hyperparameter settings.

## 3 Financial tasks for evaluation

We evaluate the performance of financial PLMs on the following six financial NLP tasks:

- FPB (Malo et al., 2014): This sentiment classification task involves three categories: positive, negative, and neutral. The dataset comprises 4,840 sentences from financial news articles.
- NER (Alvarado et al., 2015): The goal of this financial named entity recognition task is to identify four types of named entities (PER, LOC, ORG, and MISC) within financial contracts reported to the SEC.
- Headline (Sinha and Khandait, 2021): The objective of this task is to classify the impact of news articles pertaining to gold commodities based on the headline of the articles.
- FiNER (Loukas et al., 2022) This is a numeric entity recognition task. The dataset comprises XBRL-tagged financial reports from publicly traded companies.
- FinQA (Chen et al., 2021): This is a financial question-answering task for evaluating numerical reasoning and understanding. The dataset

is based on profit and loss reports of S&P 500 companies.

- FOMC (Shah et al., 2023): This aims to classify text generated by the Federal Open Market Committee (FOMC) in order to assess its impact on the financial markets. This dataset classifies the policy stance as either "Hawkish" or "Dovish." Data collection for this task was conducted up until October 15, 2022.

Figure 1b shows the embeddings of the sentences sampled from the financial tasks. The colors of each point distinguish the six tasks and the gray points represent the sentences from the pretraining data. FiNER and FinQA are located on the top side because both tasks use SEC filings to create the dataset. Meanwhile, the Headline task is located further away from the other tasks due to its focus on the gold commodity. In addition, the sentences in FPB, NER, and FOMC form their own clusters, separate from the other datasets.

As observed, each financial NLP task has unique aims and distinctive textual features. This implies that the performance of language model has to be evaluated across a broad range of tasks. For more detailed information about financial tasks, please refer to Appendix B.

## 4 Experiments

### 4.1 Experiment setup

We compared the FiLM model with existing financial domain PLMs: FinBERT-A (Araci, 2019), FinBERT-Y (Yang et al., 2020), FLANG-BERT & FLANG-RoBERTa (Shah et al., 2022), and SEC-BERT (Loukas et al., 2022). Furthermore, we fine-tune general domain PLMs, BERT (Devlin et al., 2019) and RoBERTa (Liu et al., 2019), to establish the baselines. For detailed information on each model, please refer to Appendix C.

For all experiments, except for FiNER and FinQA, the results are computed based on the average score across three seed values. For the standard deviation obtained through all experiments, please refer to Table 8.

All the models are trained on an RTX3090 GPU. For the detailed settings for fine-tuning, please refer to Appendix B. The dataset and hyperparameters used to further pretrain FiLM are provided in Appendix A. Using this setup, the training of FiLM on the entire dataset can be completed within 24 hours, resulting in a high-performance language model while maintaining a low training cost.

| Model | # Tokens (Financial Corpus) | FPB Accuracy | F-1 | NER F-1 | Headline F-1 | FiNER F-1 | FinQA Prog Acc | Exe Acc | FOMC F-1 |
|---|---|---|---|---|---|---|---|---|---|
| **General Domain PLMs** | | | | | | | | | |
| BERT (Devlin et al., 2019) | | 83.30 | 81.73 | 75.09 | 89.54 | 79.40 | 51.09 | 53.10 | 63.81 |
| RoBERTa (Liu et al., 2019) | | 85.30 | 83.93 | 78.81 | 91.29 | 81.58 | 56.76 | 59.11 | 69.16 |
| **Existing Financial Domain PLMs** | | | | | | | | | |
| FinBERT-A (Araci, 2019) | 237M | 85.25 | 82.45 | 77.93 | 90.48 | 81.49 | 47.86 | 50.04 | 64.50 |
| FinBERT-Y (Yang et al., 2020) | 4.9B | 83.68 | 82.52 | 70.40 | 90.83 | 81.08 | 38.79 | 40.54 | 64.30 |
| FLANG-BERT (Shah et al., 2022) | NA | 84.76 | 83.12 | 75.58 | 91.06 | 81.52 | 49.17 | 51.43 | 64.93 |
| FLANG-RoBERTa (Shah et al., 2022) | NA | 83.86 | 82.19 | 71.36 | 90.46 | 80.77 | 30.68 | 32.17 | 68.02 |
| SEC-BERT (Loukas et al., 2022) | 3.1B | 84.37 | 82.18 | 78.74 | 90.52 | 82.35 | 53.18 | 55.45 | 65.07 |
| **Proposed models** | | | | | | | | | |
| FiLM | 2.4B | **86.25** | **84.48** | **79.78** | **91.79** | 82.02 | 58.85 | 61.38 | **69.60** |
| FiLM (5.5B) | 5.5B | 86.14 | 84.11 | 78.82 | 91.74 | **82.39** | **59.37** | **61.64** | 69.16 |

Table 2: Main results. FiLM (5.5B) is the model trained on an additional SEC filings dataset comprising 3.1B tokens (Appendix A). In each task, the best score is marked in **bold**, while the second best score is underlined. Shah et al. (2022) did not report the number of tokens but utilized approximately 2.78 million documents.

## 4.2 Main results

Table 2 reports the results of evaluating the performance of PLMs on financial tasks. Despite further training on financial corpus, existing financial PLMs often perform worse than general domain PLMs on certain tasks. For example, RoBERTa outperforms all existing financial PLMs on FPB, FinQA, and FOMC. In addition, BERT outperforms FinBERT-A, FinBERT-Y, and FLANG-BERT on FinQA tasks. This implies that financial PLMs exhibit a deficiency in their generalization capabilities, limiting their effectiveness across a diverse spectrum of documents within the financial domain. However, our model, which incorporates a diverse range of financial corpus for pretraining, has been equipped with robust generalization capabilities, enabling it to perform well across various NLP tasks in the financial domain. Although our model is trained on fewer tokens (2.4B) than SEC-BERT (3.1B) and FinBERT-Y (4.9B), it outperforms existing PLMs on most financial tasks.

## 4.3 Impacts of the diversity in the pretraining dataset

We investigate the variation in the performance of the language model based on the number of corpus groups used. To achieve this, we generate all combinations of corpus groups and pretrain separate RoBERTa on each combination. Then, we evaluate each model by calculating the average F1 score across four downstream tasks: FPB, NER, Headline, and FiNER. Figure 2 presents the average F1 scores when the number of corpus groups varied. This indicates that as the number of groups increases, the model's performance improves. This

| Model | # Groups | NER F-1 | FiNER F-1 | FinQA Prog Acc |
|---|---|---|---|---|
| FiLM [Ours](2.4B) | 5 | **79.78** | **82.02** | **58.85** |
| w/o SEC filings(2.1B) | 4 | 76.51 | 81.94 | 57.54 |
| only SEC filings(3.1B) | 1 | 75.51 | 81.91 | 57.45 |
| only SEC filings(0.3B) | 1 | 75.30 | 81.64 | 57.10 |

Table 3: Comparison of model performance on NER, FiNER, FinQA under different pre-training data settings.

finding suggests that a more diverse corpus leads to enhanced model performance.

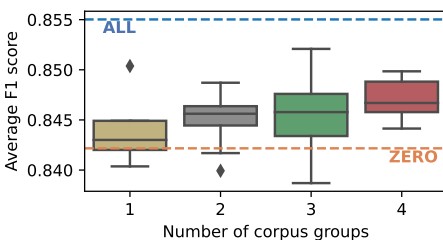

Figure 2: Average F1 scores measured on four financial tasks, with varying the number of corpus groups for pretraining.

## 4.4 Performance extrapolation

### 4.4.1 SEC filings dataset

We present empirical evidence that performance improvements can be achieved even for unseen corpus groups. To validate this, we select downstream tasks derived from SEC filings: NER, FiNER, and FinQA. Table 3 shows the performance of our model trained on a different pretraining corpus. The model pretrained on the four corpus groups without SEC filings outperform the model further trained on SEC filings only. Furthermore, despite the use of a larger number (3.1B) of tokens derived

| Model | # Tokens | # Groups | GPU(power) | GPU Time | Total Energy Consumption | NER F-1 | FiNER F-1 | FinQA Prog Acc |
|---|---|---|---|---|---|---|---|---|
| FiLM | 2.4B | 5 | RTX3090(0.36kW) | 23h | **8.3kWh** | **79.78** | **82.02** | **58.85** |
| - only SEC filings | 3.1B | 1 | RTX3090(0.36kW) | 32h | 11.2kWh | 75.51 | 81.91 | 57.45 |
| FinBERT-Y | 4.9B | 2 | Tesla P100(0.25kW) | 192h | 48.0kWh | 70.40 | 81.08 | 38.79 |

Table 4: Total energy consumptions for training FiLM and FinBERT-Y. The GPU time of FinBERT-Y is from Yang et al. (2020).

from SEC filings for further pretraining, this model does not exceed the performance of the model trained using fewer (2.1B) tokens from four different corpus groups. This demonstrates that incorporating diverse corpora can improve performance for unseen tasks, which can be more important than data volume. Furthermore, we anticipate that our model, denoted FiLM, may exhibit robust performance across a wide range of financial downstream tasks derived from unseen financial corpora.

### 4.4.2 Macroeconomic perspective

The FOMC task is a macroeconomics-based task designed to predict changes in monetary policy stance (Shah et al., 2023). This dataset comprises sentences extracted from FOMC speeches, meeting minutes, and press conferences. None of these sentences are included in pretraining dataset used for FiLM. To substantiate that FiLM performs robustly on unseen tasks, we evaluate models on the FOMC task. Table 2 represents that existing financial PLMs underperform RoBERTa, as pointed out in Shah et al. (2023). Meanwhile, our FiLM model outperforms RoBERTa. This highlights that FiLM is the first model to surpass RoBERTa in financial domain. For the results of all experiments of the FOMC task and their detailed explanation, please refer to Table 9 and Appendix D, respectively.

### 4.5 Cost-efficiency

Recently, the environmental impacts of large language models have become a critical issue due to their significant energy consumption and carbon emissions during the training model process (Strubell et al., 2019; Scao et al., 2022; Zeng et al., 2022; Touvron et al., 2023). Since further training is required for developing domain-specific PLMs, additional resources are consumed. We provide empirical evidence that when ensuring corpus diversity, energy consumption is decreased while enhancing performance. Table 4 shows the total energy consumption for training a financial PLM. We compute the electric energy required for training a model, using the following formula (Touvron et al.,

2023): Energy (Wh) = GPU power (W) × GPU time (h). Consequently, compared to FinBERT-Y, FiLM exhibits an 82% reduction in total energy consumption while achieving an average performance of 10% gain.

### Conclusion

We show that financial PLMs struggle with various financial tasks due to constrained pretraining data diversity. To address this problem, we train our FiLM model using a broad spectrum of financial data. We empirically show that FiLM outperforms other financial PLMs across six financial downstream tasks. Specifically, FiLM works robustly on unseen tasks and also attains superior performance on macroeconomics-based task. Furthermore, our experimental results indicate training on diverse corpora reduces energy consumption, resulting in environmental benefits. Our study underscores the significance of leveraging diverse data to train domain-specific language models.

### Limitations

Financial documents often attribute substantial significance to numerical data, more so than other types of documents. Therefore, we acknowledge the necessity of an efficient technique for processing numerical tokens, particularly for financial PLMs. However, when we tested several methods proposed in previous studies, we did not observe any significant improvement. Techniques for handling numeric tokens are excluded from our study; however, we highlight this as a valuable area for future investigations. Finally, our FiLM is an encoder-based model that is not suitable for generative tasks, such as financial news summarization.

### Acknowledgments

This work was supported by Institute of Information & communications Technology Planning & Evaluation(IITP) grant funded by the Korea government(MSIT) (No. RS-2023-00261068,

Development of Lightweight Multimodal Anti-Phishing Models and Split-Learning Techniques for Privacy-Preserving Anti-Phishing), (No.RS-2022-00155885, Artificial Intelligence Convergence Innovation Human Resources Development (Hanyang University ERICA)), and (2018-0-00192, the National Program for Excellence in SW). This work was supported by the National Research Foundation of Korea(NRF) grant funded by the Korea government(MSIT) (No. NRF-2022R1G1A1013549). Finally, we thank the reviewers for their detailed feedback, which helped to improve the quality of this paper.

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

# A    Pretraining information

## A.1    Datasets

- TRC2: TThe Thomson Reuters Text Research Collection (TRC2) corpus comprises 1,800,370 news stories published from 2008 to 2009. For more information on the data and acquisition, please refer to https://trec.nist.gov/data/reuters/reuters.html

- Investing.com[4]: Investing.com is a financial platform and news website that offers a wide range of information including stocks, futures, options, and commodities. The historical dataset is a compilation of news data sourced from Investing.com between 2009 and 2020.02.

- NYtimes [5]: The New York Times is a well-known newspaper worldwide. The NYTimes Dataset comprises finance-related articles collected from the NYTimes website. The data collection period spanned from 2005 to 2021.

- EIA: The U.S. Energy Information Administration (EIA) gathers, analyzes, and shares unbiased energy information to help create better policies, ensure efficient markets, and provide a public understanding of the impact of energy on the economy and environment. We collected news data only from the information provided by eia.gov.

- SEC filings[6]: The SEC receives annual reports (10-K) and quarterly reports (10-Q) from public US companies. These reports contains information about a company's business and financial performance. We downloaded the reports from SEC Edgar between 2020 and 2021. For the experiment(Section 4.4), additional data from 1993 to 2019 were included (Loukas et al., 2021).

- Earnings call: Earnings conference calls are important in delivering corporate information. Seeking Alpha[7] provides access to earnings conference call transcripts, and we collected the data.

- Arxiv[8]: This dataset is a collection of abstracts from economics research sourced from arxiv.org

- AIHUB: This dataset comprises a collection of Korean academic papers obtained and translated by professional translators. Furthermore, professors specializing in translation studies reviewed the translated corpus. We used a subset of the corpus that focused on economics-related data. This research used datasets from 'The Open AI Dataset Project (AI-Hub, South

---

[4]https://www.kaggle.com/datasets/gennadiyr/us-equities-news-data
[5]https://www.nytimes.com/section/business/economy
[6]https://www.sec.gov/
[7]https://seekingalpha.com/
[8]https://arxiv.org/search/econ

Korea)'. All data information can be accessed through 'AI-Hub'(www.aihub.or.kr).

- FinWEB[9]: The dataset is a website that provides economic knowledge and information on finance, loans, and products. We collected data by crawling websites.
- Investopedia[10]: The dataset is a financial website that functions as an economic dictionary, similar to Wikipedia, and provides definitions of economic terms. We collected all the economic terms available on the website.

### A.2 Hyperparameters for pretraining

Table 5 shows the hyperparameters used to further pretrain FiLM.

| Hyperparameters | FiLM |
|---|---|
| Initializaion | RoBERTa-base |
| Learning rate | 1e-5 |
| Batch size | 16 |
| Max epochs | 1 |
| Max length | 512 |
| Weight decay | 0 |
| Warmup steps | 0 |

Table 5: Pretraining hyperparameters setting.

### B Fine-tuning methodologies

We introduce the methods applied for finetuning PLMs on financial tasks as well as the experimental settings used. All tasks followed the methods and datasets proposed in each study. The headline task was performed using a sentiment classification method proposed by the authors[11]. Table 6 lists the descriptions, dataset sizes, and metrics for each task. We followed the finetuning method of BERT. Furthermore, classification tasks (FPB, Headline, FOMC) were finetuned using sequence classification, which involved passing the [CLS] token representation for processing. The entity recognition (NER, FiNER) task follows token classification finetuning using word representations. FinQA follows the question answering system introduced by Chen et al. (2021). Table 7 provides the parameters used to finetune each task are provided. FiNER[12] and FinQA[13] can be accessed and reviewed on the official GitHub repository provided.

---

[9]https://www.finweb.com/
[10]https://www.investopedia.com/
[11]https://www.kaggle.com/datasets/ankurzing/sentiment-analysis-in-commodity-market-gold
[12]https://github.com/nlpaueb/finer
[13]https://github.com/czyssrs/FinQA

### C Existing financial PLMs

We provide a summary of the financial PLMs used in previous research and organize the huggingface hub model used for the experiments. In the list below, items marked with "✓" indicate models that have undergone further training, while those without the mark are models trained from scratch.

- FinBERT-A ✓ (Araci, 2019) is the initial financial PLM. It is trained using the TRC2 dataset. FinBERT-A focuses on the FPB task and demonstrates superior performance compared with the original BERT. https://huggingface.co/ProsusAI/finbert

- FinBERT-Y (Yang et al., 2020) used more than three datasets compared with FinBERT-A and validated its performance across three additional tasks. In addition, instead of using the traditional BERT tokenizer and pretraining for the financial approach, FinBERT-Y generates and applies a financial vocabulary, resulting in improved performance. https://huggingface.co/yiyanghkust/finbert-tone

- FinBERT-L (Liu et al., 2021) collected a general domain corpus (3.3B tokens) and a financial corpus (12.7B tokens) to train the financial BERT model. Unlike the traditional further pretraining approach, FinBERT-L employs multitask self-supervised pretraining during the training process. However, because the proposed model is not publicly available, a comparative experiment could not be conducted.

- SEC-BERT (Loukas et al., 2022) is the first to introduce the FiNER task. The BERT model was pretrained exclusively using the SEC filings. This study emphasized constructing vocabulary solely from the financial reports dataset, with a specific focus on numerical tokens found in financial reports. In addition, SEC-BERT proposes a method for substituting numeric tokens in the context of FiNER. https://huggingface.co/nlpaueb/sec-bert-base

- FLANG-BERT & FLANG-RoBERTa ✓ (Shah et al., 2022) is the first to create a benchmark dataset for financial tasks by aggregating a diverse range of tasks. In addition,

| Name | Task | Dataset Size | | | Metric |
| | | Train | Valid | Test | |
|---|---|---|---|---|---|
| FPB | Sentiment classification | 3,391 | 726 | 726 | Accuracy & F-1 |
| NER | Named entity recognition | 932 | 232 | 302 | F-1 |
| Headline | News headlines classification | 7,989 | 1,141 | 2,282 | F-1 |
| FiNER | Numeric entity recognition | 900,384 | 112,494 | 108,378 | F-1 |
| FinQA | Question answering | 6,251 | 883 | 1,147 | Accuracy(Prog & Exe) |
| FOMC | Sentiment classification | 1,588 | 396 | 496 | F-1 (Combined-S) |

Table 6: Summary of financial tasks and their main aspects.

| Hyperparam | FPB | NER | Headline | FOMC | FiNER | FinQA |
|---|---|---|---|---|---|---|
| Learning rate | {1e-3, 1e-4, 1e-5, 2e-5, 3e-5, 4e-5, 5e-5} | | | | 1e-5 | 2e-5 |
| Batch size | {16, 32, 64} | {8, 16} | {64, 128} | {8, 16, 32} | 16 | 16 |
| Max epochs | 20 | 100 | 100 | 100 | 30 | 300 |
| Max length | {64, 128} | 512 | 64 | 256 | 200 | 512 |

Table 7: Hyperparameter settings for each downstream task.

this study investigated and applied pretraining methods optimized for economics during the finetuning process. `https://huggingface.co/SALT-NLP/FLANG-BERT`

## D Results of the FOMC task

In this section, we provide supplementary explanations for the results presented in Table 9. The FOMC task is composed of datasets that are categorized into three types:

- Meeting Minutes (MM): These are reports derived from the FOMC's eight annually scheduled meetings.
- Press Conference Transcripts (PC): These include prepared remarks as well as the Q&A sessions between the Federal Reserve Chair and the press.
- Speeches (SP): This category comprises any speeches delivered by officials of the Federal Reserve.
- Additionally, there is a "Combined" category that merges all three aforementioned types.

Texts with a "-S" indicator signify that they have been split. This is because the FOMC often employs neutral sentences to maintain market stability and minimize excessive reactions. To address this issue, the authors of this study employed a rule-based approach to split sentences that exhibit a neutral stance. All scores were obtained by strictly following the settings provided in the GitHub link[14] in the Shah et al. (2023). Numbers in parentheses indicate the standard deviation.

[14] `https://github.com/gtfintechlab/fomc-hawkish-dovish`

FiLM demonstrates superior performance across all types with the exception of "PC". When compared to RoBERTa-base, there is an improvement of +1.6 in performance in the "Combined". Notably, there are substantial increases in performance metrics for "MM-S" and "SP", with "MM-S" showing a 3.98% improvement and "SP" a 4.69% improvement.

## E Comparison with a financial LLM

Wu et al. (2023) introduced the BloombergGPT, a language model with 50 billion parameters trained on a large-scale financial corpus. Table 10 presents the results of the BloombergGPT and encoder-based PLMs on FPB and NER. For the BloombergGPT, we report the numbers provided in this study (Wu et al., 2023). Note that BloombergGPT was evaluated by conducting 5-shot learning on the FPB and 20-shot learning on the NER. Remarkably, FiLM, BERT, and RoBERTa outperformed BloombergGPT, which had many more parameters and required a higher cost. This demonstrates the limitations of using large language models for in-context learning in financial domain.

## F Visualization of financial domain datasets

To examine the distribution of sentences from financial domain datasets in the embedding space in Figure 1, we sampled 10,000 sentences from each pretraining dataset and 500 sentences from each downstream task. Then, we generate embedding representations for sampled sentences using the approach proposed in Reimers and Gurevych

| Model | FPB | | NER | Headline | FOMC |
|---|---|---|---|---|---|
| | Std(acc) | Std(F-1) | Std(F-1) | Std(F-1) | Std(F-1) |
| BERT | 1.302 | 0.682 | 1.479 | 1.852 | 1.49 |
| RoBERTa | 0.736 | 0.908 | 0.734 | 0.961 | 1.385 |
| FinBERT-A | 0.758 | 1.540 | 0.897 | 1.087 | 1.102 |
| FinBERT-Y | 1.284 | 1.014 | 1.734 | 0.927 | 1.482 |
| FLANG-BERT | 0.202 | 1.197 | 0.951 | 0.429 | 1.661 |
| FLANG-RoBERTa | 0.835 | 0.914 | 5.002 | 1.233 | 0.509 |
| SEC-BERT | 0.986 | 0.676 | 1.642 | 0.637 | 2.778 |
| FiLM | 0.605 | 0.702 | 1.528 | 1.387 | 1.802 |
| FiLM (5.5B) | 0.724 | 0.854 | 1.446 | 1.507 | 1.58 |

Table 8: The standard deviation of model performance for FPB, NER, Headline, and FOMC.

| Model | MM | MM-S | PC | PC-S | SP | SP-S | Combined | Combined-S |
|---|---|---|---|---|---|---|---|---|
| BERT | 57.82 | 63.42 | 45.33 | 53.26 | 61.93 | 61.92 | 62.60 | 63.81 |
| | (5.182) | (2.705) | (9.604) | (3.686) | (3.072) | (0.843) | (1.154) | (1.49) |
| RoBERTa | 66.96 | 66.19 | **54.08** | **54.55** | 65.08 | 66.99 | 69.04 | 69.16 |
| | (4.619) | (4.048) | (0.944) | (9.701) | (2.036) | (1.676) | (0.784) | (1.385) |
| FinBERT-A | 59.98 | 65.01 | 45.04 | 51.33 | 65.59 | 62.82 | 63.93 | 64.50 |
| | (3.786) | (2.654) | (5.62) | (5.262) | (2.341) | (3.419) | (2.625) | (1.102) |
| FinBERT-Y | 58.39 | 60.24 | 46.36 | 52.45 | 63.40 | 61.87 | 59.41 | 64.30 |
| | (3.351) | (1.093) | (8.636) | (16.101) | (0.599) | (1.175) | (3.081) | (1.482) |
| FLANG-BERT | 61.54 | 66.71 | 51.74 | 47.03 | 62.35 | 62.57 | 63.39 | 64.93 |
| | (5.214) | (1.601) | (8.891) | (7.254) | (2.973) | (0.874) | (1.261) | (1.661) |
| FLANG-RoBERTa | 62.30 | 67.18 | 51.47 | 49.44 | 65.18 | 63.53 | 64.82 | 68.02 |
| | (2.813) | (1.711) | (2.595) | (2.265) | (1.754) | (1.553) | (2.853) | (0.509) |
| SEC-BERT | 64.46 | 67.64 | 53.67 | 44.51 | 67.10 | 65.11 | 68.10 | 65.06 |
| | (8.053) | (3.455) | (2.238) | (14.825) | (2.015) | (3.114) | (0.525) | (2.778) |
| FiLM (2.4B) | **67.54** | **70.17** | 52.27 | 54.09 | **69.77** | **68.66** | **70.64** | **69.60** |
| | (4.062) | (1.682) | (3.322) | (2.2) | (2.118) | (1.901) | (1.077) | (1.802) |

Table 9: Results of the FOMC task.

| Model | Params | FPB F-1 | NER F-1 |
|---|---|---|---|
| BloombergGPT | 50B | 51.07 | 60.82 |
| BERT | 110M | 81.73 | 75.09 |
| RoBERTa | 110M | 83.93 | 78.81 |
| FiLM (2.4B tokens) | 110M | 84.48 | 79.79 |

Table 10: Results on FPB and NER for BloombergGPT, BERT, RoBERTa, and FiLM.

| | Intra-group distance |
|---|---|
| **News** | 3.529 |
| **SEC filings** | 1.819 |
| **Earnings call** | 2.557 |
| **Papers** | 2.568 |
| **MISC** | 3.378 |

Table 11: Mean distances among sentence embeddings within a corpus group.

(2019). To visualize sentence embeddings, we reduce the dimensionality of embeddings using UMAP (McInnes et al., 2018).

# G  Quantitative analysis for the corpus grouping

To confirm whether the corpus groups we divided have distinct characteristics, we visualized sentence embeddings for each group in Figure 1a. Furthermore, in this section, we aim to establish the unique features of each group through quantitative analysis. We calculate the mean distances between sentence embeddings within a corpus group (Table 11) and across corpus groups (Table 12). Inter-group distances are generally larger than intra-group distances. Thus, sentences within the same group have certain similarities. This result supports our claim that creating training data using diverse corpus groups is a more effective approach. This is because the distinct characteristics of each group contribute to a more comprehensive and varied learning experience.

# H  Effectiveness of the deduplication

We compared the results before and after applying the preprocessing step to the proposed method (Section 2.2). Specifically, we compare the model's performance trained on the entire corpus before pre-

|  |  | News | SEC filings | Earnings call | Papers | MISC |
|---|---|---|---|---|---|---|
| **Inter-group distance** | **News** | — | 5.062 | 3.574 | 3.542 | 4.933 |
|  | **SEC filings** | 5.062 | — | 3.832 | 4.763 | 3.910 |
|  | **Earnings call** | 3.754 | 3.832 | — | 2.917 | 3.518 |
|  | **Papers** | 3.542 | 4.763 | 2.917 | — | 4.008 |
|  | **MISC** | 4.933 | 3.910 | 3.518 | 4.008 | — |

Table 12: Mean distances among sentence embeddings across corpus groups.

| Model | # Tokens | FPB | | NER | Headline | FiNER | FinQA | | FOMC |
|---|---|---|---|---|---|---|---|---|---|
|  | (Financial Corpus) | Accuracy | F-1 | F-1 | F-1 | F-1 | Prog Acc | Exe Acc | F-1 |
| Before | 3.3B | **86.70** | 83.38 | 79.16 | 89.74 | 81.95 | 58.50 | 60.68 | 67.89 |
| FiLM[After] | 2.4B | 86.25 | **84.48** | **79.78** | **91.79** | **82.02** | **58.85** | **61.38** | **69.60** |

Table 13: Compare the performance of before and after preprocessing models.

processing, referred to as the 'Before' model, with the performance of our FiLM model trained using the proposed method after preprocessing. Table 13 presents the results comparison. In addition, we compare the number of duplicate sentences before and after preprocessing for each dataset, as shown in Table 14.

| Name | Original sentences | Duplicate sentences | Duplicate ratio |
|---|---|---|---|
| SEC filings | 13,969,168 | 5,565,936 | 39.84 |
| Earnings call | 77,864,753 | 1,789,634 | 2.30 |
| TRC2 | 7,784,537 | 1,297,580 | 16.67 |
| NYtimes | 3,424,448 | 69,926 | 2.04 |
| Papers | 1,777,151 | 8,589 | 0.48 |
| EIA | 45,496 | 1,966 | 4.32 |
| Investing.com | 220,890 | 423 | 0.19 |
| FinWEB | 144,456 | 390 | 0.27 |
| AIHUB | 278,054 | 6 | 0.00 |
| Investopedia | 238,344 | 0 | 0.00 |
| Total | 105,747,297 | 8,734,450 | 8.26 |

Table 14: Comparison of original sentences and duplicate sentences.

## I  Similarity between corpus groups and downstream tasks

In this section, we conduct a qualitative analysis using corpus embeddings to measure the similarity between corpus groups and tasks. The Figure 1a embedding map illustrates that even within the same Financial Domain Dataset, each corpus has distinct characteristics, leading to a wide distribution in the embedding space. Furthermore, to confirm similar corpus groups for each downstream task in financial domain, we calculate both the vocabulary overlap ratio and distances in the embedding space. Figure 3 shows the vocabulary ratio between the corpus groups and downstream tasks. The top 1,000 most frequent unigrams are used to calculate the ratio. Figure 4 shows the two

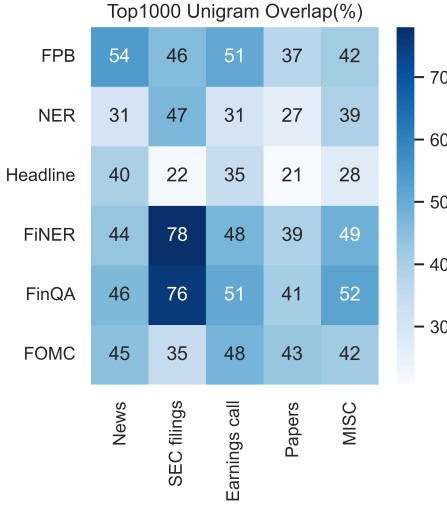

Figure 3: Vocabulary overlap ratio between pretraining and downstream task datasets.

closest corpus groups for each downstream task in the embedding space. ● markers represent sentence embeddings in corpus groups, while ✕ markers indicate sentence embeddings in downstream task. The mean distance of sentence embeddings between each corpus group and downstream task are calculated, and the two nearest groups are selected based on these distances. We discover that there are slight differences between corpus groups identified as similar in the embedding space and those identified as similar based on the vocabulary overlap ratio. For instance, we noted that the News corpus group demonstrated similarity to FPB in vocabulary overlap, but the SEC-filings corpus group showed similarity to FPB within the embedding space. This indicates that multiple factors should be considered comprehensively when assessing the similarity between the datasets.

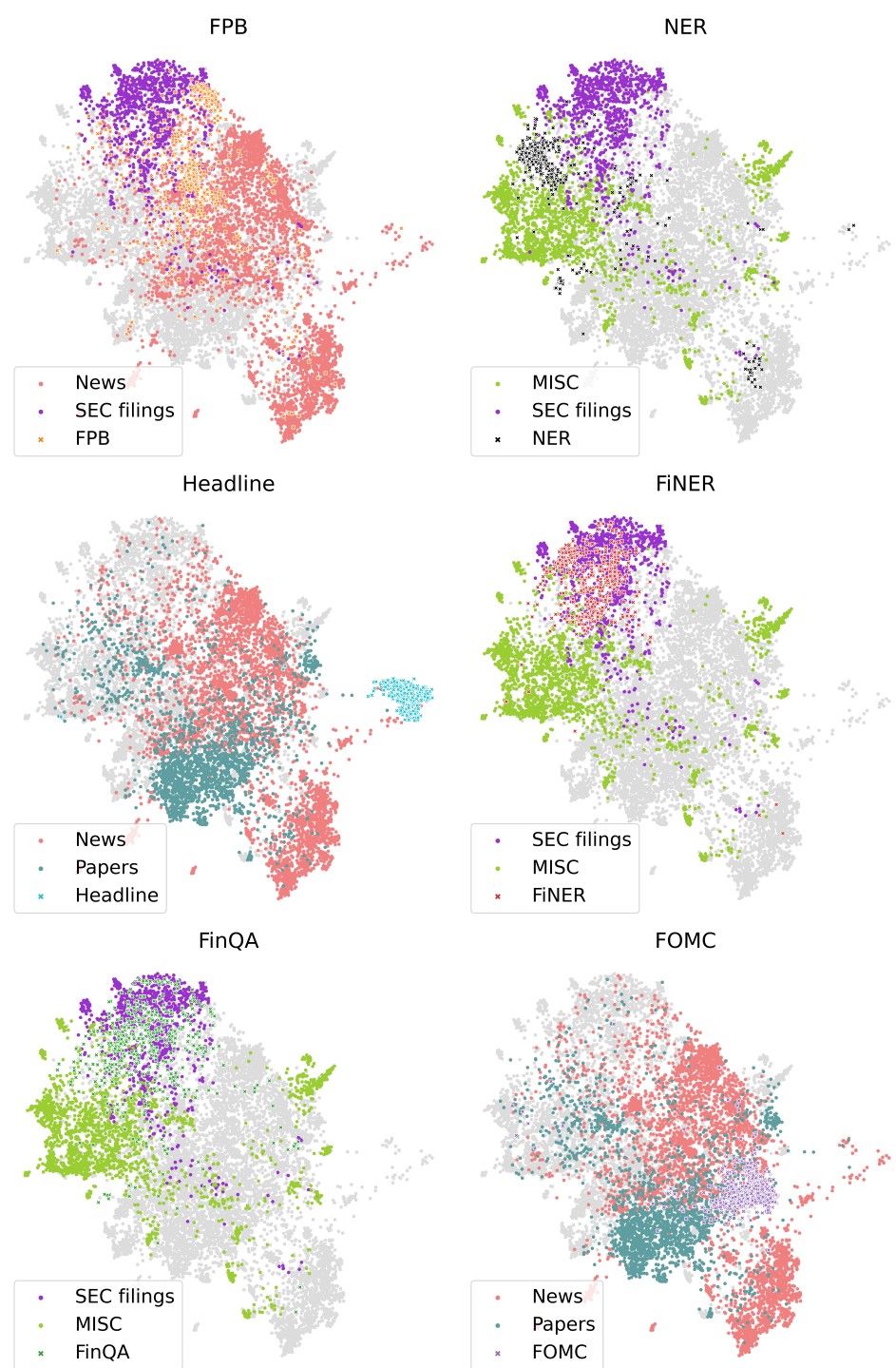

Figure 4: Two nearest corpus groups to each downstream dataset in embedding space.