# OpenReview forum: "Exploring the Impact of Corpus Diversity on Financial Pretrained Language Models"
_EMNLP/2023/Conference — EMNLP 2023 Findings_

### Official Review · Reviewer_ptSf · 2023-08-04

**Soundness:** 3

**Excitement:**

4: Strong: This paper deepens the understanding of some phenomenon or lowers the barriers to an existing research direction.

**Paper Topic And Main Contributions:**

The authors observed that the existing pretrained language models in the finance domain are pretrained on less diverse finance data, and hence the authors investigated whether pretraining on diverse finance data results in better models. The authors pretrained the FiLM model (initialized from the RoBERTa model) using finance texts gathered from diverse sources and showed that FiLM outperformed existing models. Because of the diversity in the data, the FiLM model pretrained using just 2.4B tokens  outperformed the existing models pretrained using more than 3B tokens.

**Questions For The Authors:**

- Most of the existing finance pretrained language models are either pretrained from scratch using BERT model architecture or pretrained after initializing from the BERT model. Why did the authors choose to initialize the model from the RoBERTa model instead of the BERT model? RoBERTa model is pretrained on 160GB of data, while the BERT model is pretrained on just 16B of data. So,initializing from the RoBERTa model will give an additional advantage to the FiLM model compared to the existing models, which are initialized from the BERT model.
- In Table 2, the authors presented the results for FiLM model pretrained using 2.4B and 5.5B tokens. It is never mentioned in the paper, how 5.5B token corpus is obtained.
- I suggest the authors to include experiments using FLANG-RoBERTa to have a more fair comparison. Here is the link for the model - https://huggingface.co/SALT-NLP/FLANG-Roberta



**Reasons To Accept:**

- The paper is well-written, easy to understand, and the findings are well supported by the experiment results.
- The paper highlights the importance of having more diversified finance data in the pretraining corpus to get better models.
- The FiLM model pretrained using just 2.4B tokens from diverse sources outperformed existing models pretrained using a less diversified corpus having more than 3B tokens.


**Reasons To Reject:**

- The performance of FiLM is compared with existing models, which are either pretrained from scratch using BERT architecture or pretrained after initializing from the BERT model. Here the main concern is the initialization of FiLM from the RoBERTa model instead of the BERT model.

**Reproducibility:**

4: Could mostly reproduce the results, but there may be some variation because of sample variance or minor variations in their interpretation of the protocol or method.

**Reviewer Confidence:**

5: Positive that my evaluation is correct. I read the paper very carefully and I am very familiar with related work.

---

> ### Author Rebuttal · Authors · 2023-08-29
>
> Thank you for your review and valuable feedback!
>
> > Q1. Most of the existing finance pretrained language models are either pretrained from scratch using BERT model architecture or pretrained after initializing from the BERT model. Why did the authors choose to initialize the model from the RoBERTa model instead of the BERT model? RoBERTa model is pretrained on 160GB of data, while the BERT model is pretrained on just 16B of data. So,initializing from the RoBERTa model will give an additional advantage to the FiLM model compared to the existing models, which are initialized from the BERT model.
> >
>
> > Q3. I suggest the authors to include experiments using FLANG-RoBERTa to have a more fair comparison. Here is the link for the model - https://huggingface.co/SALT-NLP/FLANG-Roberta
> >
>
> Thanks for the valuable feedback to improve our experiments. Following the reviewer’s suggestion, we conducted experiments with the FLANG-RoBERTa model. The proposed model FiLM outperform FLANG-RoBERTa in all settings. The result demonstrate the effectiveness our method.
>
> We will include the experimental results of FLANG-RoBERTa in our revised paper.
>
> | Model | FPB |  | NER | Headline | FiNER | FinQA |  |
> | --- | --- | --- | --- | --- | --- | --- | --- |
> | Metric | Accuracy | F-1 | F-1 | F-1 | F-1 | Prog Acc | Exe Acc |
> | FLANG-BERT | 86.18 | 83.85 | 76.35 | 93.20 | 81.52 | 49.17 | 51.43 |
> | FLANG-RoBERTa | 85.78 | 83.14 | 71.21 | 93.41 | 80.77 | 30.68 | 32.17 |
> | FiLM | 87.17  | 85.32  | 81.63  | 93.69  | 82.02  | 58.85 | 61.38 |
> | FiLM(5.5B) | 88.06 | 84.68 | 80.67 | 93.86 | 82.39 | 59.37 | 61.64 |
>
> > Q2. In Table 2, the authors presented the results for FiLM model pretrained using 2.4B and 5.5B tokens. It is never mentioned in the paper, how 5.5B token corpus is obtained.
> >
>
> Thank you for identifying an oversight in our paper.
>
> To obtain the 5.5B token corpus, we use additional SEC filings data containing 3.1B tokens. Note that the additional data is also used in the experiment in Section 4.4. We will specify it in our revised paper.

---

### Official Review · Reviewer_7icf · 2023-08-11

**Soundness:** 3

**Excitement:**

2: Mediocre: This paper makes marginal contributions (vs non-contemporaneous work), so I would rather not see it in the conference.

**Paper Topic And Main Contributions:**

The paper discusses a financial language model called FiLM, abbreviated for Financial Language Model. The main research question for this work revolves around the premise that the financial literature data can be split into 5 major groups and most of the Pretrained Language Models (PLMs) further trained on financial data have only 2 out of 5 groups in their training corpus, as showed by multiple examples in the paper.

The main research question framed is whether a model trained on a more diverse training data would perform better then the existing PLMs. The author thoroughly explore the research question by curating a dataset which contains all of the 5 different data groups and further training a masked language model (RoBERTa) on a pre-processed version of the data.

They compare their FiLM model across 5 different tasks against 4 different existing PLMs and 2 baselines finetuned for each specific task. The results show that the FiLM model outperforms each of the 6 (4 PLMs + 2 baselines) models in all of the tasks. They also present an analysis of the effect of number of groups on the evaluation across an average F1 score across 4 different tasks.

The main contributions of the paper are:

* Analysis of different pre-existing financial PLMs and the shortcomings in terms of data diversity in their datasets.

* A new dataset curated which has a higher diversity across 5 different data groups for financial data

* FilLM models trained on this curated data which outperforms even BloombergGPT which is exponentially bigger and trained on a much larger data corpus

* An analysis of effect of number of groups included in the training dataset on different evaluation metrics.

**Questions For The Authors:**

A. Was there any analysis done on general out-of-domain tasks WiC, GLUE or MNLI?

B. Can you add the state-of-the-art models evaluations for each of the tasks in table 2?

C. What is the FiLM (5.5B) model mentioned in table 2, which showcases best performance in almost all of the tasks? I could not find reference to that specific training data which contains 5.5B tokens?

**Reasons To Accept:**

* The work presents strong evidence in favor of inclusion of a more diverse dataset in model training

* The work showcases better performance than in-domain LLM over domain-specific tasks

**Reasons To Reject:**

* The paper mainly focuses on domain-specific training/evaluation and fails to give reason as to why this is better than a pre-existing generalized LLM

* Even though the work showcases strong performance on in-domain tasks, the paper fails to report the performance comparison/effect on general out-of-domain tasks

* Though the work presented has stronger contributions to the financial community, the work fails to give a novel contribution to the NLP community. The work presents the improvements of PLM trained on a more diverse data, which is a common knowledge.

**Reproducibility:**

4: Could mostly reproduce the results, but there may be some variation because of sample variance or minor variations in their interpretation of the protocol or method.

**Reviewer Confidence:**

3: Pretty sure, but there's a chance I missed something. Although I have a good feel for this area in general, I did not carefully check the paper's details, e.g., the math, experimental design, or novelty.

---

> ### Author Rebuttal · Authors · 2023-08-29
>
> Thank you for your review and valuable feedback!
>
> > The paper mainly focuses on domain-specific training/evaluation and fails to give reason as to why this is better than a pre-existing generalized LLM
> >
>
> The underperformance of general-domain LLM arises from the distinct textual characteristics between general-domain and financial corpora. For instance, financial reports prominently include numerical information(Section 2.1.). To substantiate this claim, we compute the mean distances among sentence embeddings within and across domains. Consequently, the inter-domain distance are larger than the intra-domain distances. This difference in distance indicates dissimilarity between general-domain and financial corpora. Nevertheless, existing financial PLMs exhibit subpar performance on many financial tasks, even when compared to general domain PLMs, due to the deficiency of corpus diversity. However, our FiLM models outperform not only financial PLMs but also surpass general-domain PLMs in financial tasks.
>
> |  | Intra-domain |  | Inter-domain |
> | --- | --- | --- | --- |
> |  | Financial | General | Financial-General |
> | Avg distance between sentences | 3.7205 | 4.2048 | 6.2576 |
>
> > Though the work presented has stronger contributions to the financial community, the work fails to give a novel contribution to the NLP community. The work presents the improvements of PLM trained on more diverse data, which is common knowledge.
> >
>
> While it is widely accepted that using a diverse corpus is important for improving generalization performance, most domain-specific studies focused on collecting a large corpus and overlooked the importance of corpus diversity. Below, We’d like to emphasize the unique and important aspects of our paper:
> 1. **Cost-efficiency**
>
>     Compared to FinBERT-Y, our FiLM model exhibits an 88% decrease in GPU time, while achieving an average performance increase of 10%. Our study empirically demonstrates that harnessing diverse corpus groups enhances model performance while reducing the number of tokens, GPU time, and energy consumption.
>
>     | Model | # Tokens | # Corpus Groups | GPU (power) | GPU Time | Total Energy Consumption  | NER(F1) | FiNER(F1) | FinQA(F1) |
>     | --- | --- | --- | --- | --- | --- | --- | --- | --- |
>     | FiLM | 2.4B | 5 | RTX3090 (0.36kW) | 23h | 8.3kWh | 81.63 | 82.02 | 58.85 |
>     | FinBERT-Y | 4.9B | 2 | Tesla P100 (0.25kW) | 192h | 48.0kWh | 72.38 | 81.08 | 38.79 |
> 2. **Beyond data volume: corpus diversity as a significance factor for generalization performance**
>
>     Even with 1 billion fewer tokens trained, the model trained on the remaining 4 corpus groups, excluding SEC-filings, exhibits superior performance compared to the model exclusively trained on SEC filings, when evaluating them on NER, FiNER, and FinQA tasks derived from SEC-filings. (Table 3) This result emphasizes that corpus diversity plays a significant role in the generalization performance of the model, beyond the data volume.
>
>     | Model | # Tokens | # Corpus Groups | GPU (power) | GPU Time | Total Energy Consumption  | NER(F1) | FiNER(F1) | FinQA(F1) |
>     | --- | --- | --- | --- | --- | --- | --- | --- | --- |
>     | FiLM w/o SEC Filings | 2.1B | 4 | RTX3090 (0.36kW) | 20h | 7.2kWh | 80.55 | 81.94 | 57.54 |
>     | FiLM only SEC Filings | 3.1B | 1 | RTX3090 (0.36kW) | 32h | 19.8kWh | 79.76 | 81.91 | 57.45 |
>
> In summary, the importance of corpus diversity has been neglected in previous PLM works. Our study highlights that leveraging diverse corpus groups not only boosts model performance but also significantly (more than 80%) cuts down on GPU time, and energy consumption.
>
> > Can you add the state-of-the-art models evaluations for each of the tasks in table 2?
> >
>
> SEC-BERT is the state-of-the-art model for NER, FiNER and FinQA tasks, while FLANG-BERT exhibits superior performance in other financial tasks. In conclusion, our FiLM model outperforms the-state-of-the-art model in each financial task (Table 2).
>
> > What is the FiLM (5.5B) model mentioned in table 2, which showcases the best performance in almost all of the tasks? I could not find reference to that specific training data which contains 5.5B tokens?
> >
> Thank you for identifying an oversight in our paper.
>
> To obtain the 5.5B token corpus, we use additional SEC filings data containing 3.1B tokens. Note that the additional data is also used in the experiment in Section 4.4. We will specify it in our revised paper.
>
> > Was there any analysis done on general out-of-domain tasks WiC, GLUE or MNLI?
> >
>
> Since our study focuses on domain-specific PLMs, the model evaluations have been concentrated on financial tasks. We are preparing the experiments on general domain tasks, however, we cannot include the result due to time constraints. We will report the performance of models on out-of-domain tasks in the camera-ready version of the paper.

---

### Official Review · Reviewer_dzK5 · 2023-08-12

**Soundness:** 4

**Excitement:**

4: Strong: This paper deepens the understanding of some phenomenon or lowers the barriers to an existing research direction.

**Paper Topic And Main Contributions:**

In this study the authors develop FiML, a RoBERTA model adapted to the financial domain through additional pre-training of specialized corpus, and test its performance in comparison to other general purpose and specialized models on a diverse set of tasks.
The main contribution of the paper include the development and characterization of domain-specific training dataset, as well as the development of a publicly available financial-domain model, FiML, which outperforms baseline models on several downstream tasks.

**Reasons To Accept:**

This study contributes significantly to the field by introducing and making a new financial language model available to the public. Moreover the authors characterize the training dataset providing insights regarding the importance of input diversity for efficient pre-training and better model generalizability. The paper is very clear and well written.

**Reasons To Reject:**

It is unclear wether the other specialized (financial) LM have been trained from scratch or domain adaptions of general purpose models as is the case of FiML.

**Reproducibility:**

5: Could easily reproduce the results.

**Reviewer Confidence:**

2: Willing to defend my evaluation, but it is fairly likely that I missed some details, didn't understand some central points, or can't be sure about the novelty of the work.

---

> ### Author Rebuttal · Authors · 2023-08-29
>
> Thank you for your review and valuable feedback!
>
> > It is unclear wether the other specialized (financial) LM have been trained from scratch or domain adaptions of general purpose models as is the case of FiML.
> >
>
> FinBERT-Y and SEC-BERT are trained from scratch, while FinBERT-A and FLANG-BERT are domain-adaptive PLMs as is the case of our FiLM models.
>
> We will include this information in our paper.

---

### Official Review · Reviewer_8cJY · 2023-08-12

**Soundness:** 3

**Excitement:**

3: Ambivalent: It has merits (e.g., it reports state-of-the-art results, the idea is nice), but there are key weaknesses (e.g., it describes incremental work), and it can significantly benefit from another round of revision. However, I won't object to accepting it if my co-reviewers champion it.

**Paper Topic And Main Contributions:**

The paper investigates the impact of diversity in the pre-training data for foundational models trained for financial domain. The authors show that more diversity leads to better overall performance across a variety of tasks.

**Questions For The Authors:**

Please refer to the weaknesses section.

**Reasons To Accept:**

- Foundational models pre-trained for specific domains often fall short in performance compared to their open-domain counterparts. Identifying the issues and potential solutions to address this challenge is crucial. The authors of this paper have centered their focus on this significant problem and constructed an analysis around it.

**Reasons To Reject:**

1. The analysis lacks essential details. The experimental setup's clarity, like the variance and mean of fine-tuning performance across tasks, and the rationale behind "# of tokens NA" for FLANG-BERT, remains incomplete.

2. For the FLANG-<model> family, a RoBERTa version of the model also exists. Comparing a RoBERTa-based FLANG with FiLM would provide a more equitable comparison.

3. The paper concludes that enhanced corpus pre-training diversity leads to improved results across tasks, a widely accepted notion in the NLP community, substantiated by numerous research papers. Therefore, the presented results lack novelty unless I've overlooked a crucial aspect of the paper.

4. The paper lacks a qualitative error analysis demonstrating why FiLM outperforms other models on the issues mentioned in the introduction.

**Reproducibility:**

4: Could mostly reproduce the results, but there may be some variation because of sample variance or minor variations in their interpretation of the protocol or method.

**Reviewer Confidence:**

5: Positive that my evaluation is correct. I read the paper very carefully and I am very familiar with related work.

**Typos Grammar Style And Presentation Improvements:**

I believe a proofreading would improve the paper's overall presentation.
1. There are a lot of details which are not quite important for the main section of the paper, such as, the discussion around the corpus's embeddings.
2. In the abstract, acronym FiLM is used while it has not been introduced before.
3. Taking an example of the first sentence of the conclusion - it could be worded better and concise as "We show that financial PLMs struggle with diverse financial tasks due to constrained pretraining data diversity". Similarly, there are a lot of improvements that can be made throughout the paper.

---

> ### Author Rebuttal · Authors · 2023-08-29
>
> Thank you for your thorough and detailed comments. We have carefully addressed all the points you raised.
>
> > The analysis lacks essential details. The experimental setup's clarity, like the variance and mean of fine-tuning performance across tasks, and the rationale behind "# of tokens NA" for FLANG-BERT, remains incomplete.
> >
>
> Our experimental setup was primarily based on the FLUE benchmark (Shah, Raj, et al. 2022)[1]. However, for tasks that were not included in the FLUE benchmark, such as FiNER and FinQA, we adhered to the settings in the respective papers and their GitHub repositories.
>
> Regarding the section labeled "# of tokens NA," we marked it as NA in Table 1 of our paper because the corpus used in Shah, Raj, et al. (2022) records the number of documents instead of the number of tokens. However, as you have pointed out, we will replace NA with a "# of documents" notation.
>
> We will clarify the experimental setups in the revised paper.
>
> > For the FLANG-<model> family, a RoBERTa version of the model also exists. Comparing a RoBERTa-based FLANG with FiLM would provide a more equitable comparison.
> >
>
> Thanks for the valuable feedback to improve our experiments. Following the reviewer’s suggestion, we conducted experiments with the FLANG-RoBERTa model. The proposed model FiLM outperforms FLANG-RoBERTa in all settings. The result demonstrates the effectiveness of our method.
>
> We will include the experimental results of FLANG-RoBERTa in our revised paper.
>
> | Model | FPB |  | NER | Headline | FiNER | FinQA |  |
> | --- | --- | --- | --- | --- | --- | --- | --- |
> | Metric | Accuracy | F-1 | F-1 | F-1 | F-1 | Prog Acc | Exe Acc |
> | FLANG-BERT | 86.18 | 83.85 | 76.35 | 93.20 | 81.52 | 49.17 | 51.43 |
> | FLANG-RoBERTa | 85.78 | 83.14 | 71.21 | 93.41 | 80.77 | 30.68 | 32.17 |
> | FiLM | 87.17  | 85.32  | 81.63  | 93.69  | 82.02  | 58.85 | 61.38 |
> | FiLM(5.5B) | 88.06 | 84.68 | 80.67 | 93.86 | 82.39 | 59.37 | 61.64 |
>
> > The paper concludes that enhanced corpus pre-training diversity leads to improved results across tasks, a widely accepted notion in the NLP community, substantiated by numerous research papers. Therefore, the presented results lack novelty unless I've overlooked a crucial aspect of the paper.
> >
>
> While it is widely accepted that using a diverse corpus is important for improving generalization performance, most domain-specific studies focused on collecting a large corpus and overlooked the importance of corpus diversity. Below, We’d like to emphasize the unique and important aspects of our paper:
>
> 1. **Cost-efficiency**
>
>     Compared to FinBERT-Y, our FiLM model exhibits an 88% decrease in GPU time, while achieving an average performance increase of 10%. Our study empirically demonstrates that harnessing diverse corpus groups enhances model performance while reducing the number of tokens, GPU time, and energy consumption.
>
>     | Model | # Tokens | # Corpus Groups | GPU (power) | GPU Time | Total Energy Consumption  | NER(F1) | FiNER(F1) | FinQA(F1) |
>     | --- | --- | --- | --- | --- | --- | --- | --- | --- |
>     | FiLM | 2.4B | 5 | RTX3090 (0.36kW) | 23h | 8.3kWh | 81.63 | 82.02 | 58.85 |
>     | FinBERT-Y | 4.9B | 2 | Tesla P100 (0.25kW) | 192h | 48.0kWh | 72.38 | 81.08 | 38.79 |
> 2. **Beyond data volume: corpus diversity as a significance factor for generalization performance**
>
>     Even with 1 billion fewer tokens trained, the model trained on the remaining 4 corpus groups, excluding SEC-filings, exhibits superior performance compared to the model exclusively trained on SEC filings, when evaluating them on NER, FiNER, and FinQA tasks derived from SEC-filings. (Table 3) This result emphasizes that corpus diversity plays a significant role in the generalization performance of the model, beyond the data volume.
>
>     | Model | # Tokens | # Corpus Groups | GPU (power) | GPU Time | Total Energy Consumption  | NER(F1) | FiNER(F1) | FinQA(F1) |
>     | --- | --- | --- | --- | --- | --- | --- | --- | --- |
>     | FiLM w/o SEC Filings | 2.1B | 4 | RTX3090 (0.36kW) | 20h | 7.2kWh | 80.55 | 81.94 | 57.54 |
>     | FiLM only SEC Filings | 3.1B | 1 | RTX3090 (0.36kW) | 32h | 19.8kWh | 79.76 | 81.91 | 57.45 |
> 3. **The first PLM to surpass RoBERTa in various financial tasks**
>
>     As pointed out in a recent work Shah et el. (2023)[2],  there has been no financial PLM that outperforms RoBERTa in various financial tasks. Our FiLM model is the first model to surpass RoBERTa in the financial domain. To substantiate this claim, we additionally evaluate our FiLM model on FOMC task(Shah et al., 2023 [2]). The result shows superior performance of our model compared to existing financial PLMs. We will report these results in the camera-ready version of our paper.
>
>     | Model | FOMC (F1-score) |
>     | --- | --- |
>     | FinBERT-A | 0.6325 |
>     | FLANG-BERT | 0.6307 |
>     | FLANG-RoBERTa | 0.6482 |
>     | RoBERTa-base | 0.6755 |
>     | FiLM | 0.7064 |
>
> In summary, the importance of corpus diversity has been neglected in previous PLM works. Our study highlights that leveraging diverse corpus groups not only boosts model performance but also significantly (more than 80%) cuts down on GPU time, and energy consumption.
>
> > The paper lacks a qualitative error analysis demonstrating why FiLM outperforms other models on the issues mentioned in the introduction.
> >
>
> Our qualitative analysis is conveyed through Figure 1, which focuses on Corpus Embeddings. The embedding map illustrates that even within the same Financial Domain Dataset, each corpus has distinct characteristics, leading to a wide distribution in the embedding space. For a deeper understanding of the relationship between the tasks and the corpora, we recommend referring to Figure 4 in Appendix Section H.
> Furthermore, as the reviewer suggested, we will enhance qualitative error analysis. Specifically, we are going to conduct a detailed analysis of instances where FiLM made errors in various tasks, examining both the model's limitations and potential areas for improvement. Additionally, we will analyze cases where FiLM correctly answered questions that other models got wrong. Through this, we aim to clearly elucidate the unique strengths of the FiLM model.
>
> [1] Shah, Raj, et al. "When FLUE Meets FLANG: Benchmarks and Large Pretrained Language Model for Financial Domain." *Proceedings of the 2022 Conference on Empirical Methods in Natural Language Processing*. 2022.
>
> [2] Shah, Agam, Suvan Paturi, and Sudheer Chava. "Trillion Dollar Words: A New Financial Dataset, Task & Market Analysis." *Proceedings of the 61st Annual Meeting of the Association for Computational Linguistics (ACL)*. 2023. [https://arxiv.org/abs/2305.07972](https://arxiv.org/abs/2305.07972)

---

### Meta-Review · Area_Chair_LMLN · 2023-09-18

**Recommendation:** 3

**Metareview:**

A new LLM is trained on a diverse set of financial datasets and obtains improved performance on target financial related NLP tasks. Reviewers overall appreciated the improved empirical results on financial datasets and the release of  a new larger LLM on this domain. However, there was less excitement due to the very focused scope (only financial domain) and the lack of novelty in the method or conclusions (helpfulness of diversity is already known).

In terms of soundess, overall most reviewers found the evaluation solid. However, it can definitely be improved. For instance, figure 1 is unclear (sides a and b look identical only with/without colors?... and overall this 2D visualization is more suitable as an interpretability tool but is not a proof to motivate a method on top). Also, reporting confidence intervals would be better (e.g. with bootsrapping). Reporting out of domain results would be helpful. Table 2 is not explained (I assume underline is for second best?)

---

### Decision · Program_Chairs · 2023-10-07

**Decision:**

Accept-Findings

**Comment:**

A new LLM is trained on a diverse set of financial datasets and obtains improved performance on target financial related NLP tasks. Reviewers overall appreciated the improved empirical results on financial datasets and the release of  a new larger LLM on this domain. However, there was less excitement due to the very focused scope (only financial domain) and the lack of novelty in the method or conclusions (helpfulness of diversity is already known).

In terms of soundess, overall most reviewers found the evaluation solid. However, it can definitely be improved. For instance, figure 1 is unclear (sides a and b look identical only with/without colors?... and overall this 2D visualization is more suitable as an interpretability tool but is not a proof to motivate a method on top). Also, reporting confidence intervals would be better (e.g. with bootsrapping). Reporting out of domain results would be helpful. Table 2 is not explained (I assume underline is for second best?)